# Evaluation of the Efficiency of Hand Hygiene Technique with Hydroalcoholic Solution by Image Color Summarize

**DOI:** 10.3390/medicina58081108

**Published:** 2022-08-16

**Authors:** Catalina Iulia Saveanu, Daniela Anistoroaei, Stefan Todireasa, Alexandra Ecaterina Saveanu, Livia Ionela Bobu, Irina Bamboi, Octavian Boronia, Carina Balcos

**Affiliations:** Faculty of Dental Medicine, Grigore T. Popa University of Medicine and Pharmacy, 700115 Iasi, Romania

**Keywords:** hand hygiene, hydroalcoholic-solution, fluorescence, control infection

## Abstract

*Background and Objectives*: The HH (hand hygiene) technique is relatively simple. Even so, in practice there are still non-conformities regarding this aspect. Lack of knowledge or lack of compliance can be reasons for non-adherence with HH techniques. In this context, the purpose of this study was to follow the realization of the hand-washing technique with hydroalcoholic solution, applied before and after receiving the HH recommendations. *Materials and Methods*: A descriptive, cross-sectional study was conducted from May 2022 to June 2022. Students from a second year dental medicine class teaching in French 2021/22 “Grigore T. Popa” University of Medicine and Pharmacy in Iasi were included in the study. The approval of the ethics commission was received: No. 184/05.05.2022. The study was conducted in two stages. In the first stage, HH was performed without any recommendation. In the second stage, the antiseptic hand rubbing technique was presented following the WHO recommendations. The fluorescent Fluo-Add solution, Wood lamp for dermatology (4 × 5.5 W ultraviolet tubes with a wavelength of 360 nm), and photo camera were used. Subjects performed their HH movement before and after receiving instructions according to WHO. Images were initially taken from the backhand and palm and were finally analyzed with Image Color Summarizer. The data were analyzed by the Mann–Whitney U Test, *t*-test paired samples using IBM-SPSS version 26 (IBM, Armonk, NY, USA), and *p* ≤ 0.05 was considered statistically significant. *Results*: After analyzing the images, there were 70 subjects, 45.7% (32) female and 54.33% (38) male. Final average covered area of backhand was 60.89% (±17.17), 28.84:86.11, compared to 52.07% (±17.04), 9:85.23. Final average covered area for palm was 69.91% (±13.5), 31.61:93.41 compared to 59.74% (±16.64), 26.13:92.72. No statistical significance was obtained by gender. *Conclusions*: The study showed an improvement in hand hygiene technique without highlighting gender differences.

## 1. Introduction

Healthcare-associated infections (HCAIs) are a major problem for patients’ safety as they may lead to prolonged hospital stays, long-term disability, increased antibiotic resistance, high costs for patients and their families, and excess deaths [1]. It is estimated that 5–10% of patients admitted to acute care hospitals in developed countries acquire HCAIs [2]. In particular, developing countries are at high risk of HCAIs because of unfavorable factors, such as understaffing, poor hygiene and sanitation, lack or shortage of basic equipment, and overcrowding [1,2]. Two types of microorganisms comprise the normal flora of hands: transient flora and resident flora [3]. Transient flora, which is often acquired by HCP during contact with patients or environmental surfaces, colonize the superficial layers of the skin. The transient flora represents the microorganisms most commonly associated with HAIs and are easier to remove from the skin by using an ABHS or washing hands.

Hand hygiene (HH) is a general term referring to any action of hand cleansing involving HW, antiseptic HW, antiseptic hand rubbing, or surgical hand antisepsis. HW is a “procedure of washing hands with plain or antimicrobial soap and water”; it can be performed by means of different methods, depending on the risk of infection from healthcare activities.

According to the WHO recommendations, alcohol-based hand rubs are the only known means for rapidly and effectively inactivating a wide array of potentially harmful microorganisms on hands [1]. WHO recommends alcohol-based hand rubs based on the following factors: evidence-based, fast-acting, and broad-spectrum microbicidal activity; suitability for use in resource-limited or remote areas; capacity to promote improved compliance with HH; economic benefit; minimization of risks from adverse events [1].

Guidelines for HH prepared by various other agencies are currently available. Both WHO and CDC guidelines are documents prepared specifically to promote HH. Hand hygiene is considered a simple and effective measure to reduce HCAIs across all the healthcare settings [3]. It is assumed that simple HW with liquid soap could save 1 million lives a year, and many public health campaigns worldwide have addressed “HH” with varying success [4].

The other common method of HH is the use of high-quality hand disinfectant products, such as alcohol-based hand sanitizers (ABHSs) that do not require the use of water. ABHSs contain additional skin care substances, such as emollients and humectants, which help replace some of the water that is stripped by the alcohol [5]. The Centers for Disease Control and Prevention (CDC) guidelines on environmental control and HW published in 1985 recommended that alcohol-containing solutions only be used for HH in emergency settings where sinks were not available [6]. Use of ABHSSs temporarily reduces the number of microbiotas on the hands without producing significant long-term changes in the hand microbiome [7,8,9,10,11,12,13].

Following publication of the CDC and WHO guidelines, most efforts to promote improved HH have focused on increasing HH compliance with little attention paid to how HH is performed (HH technique) [1]. As an example, several studies have reported that high-adherence rates were accompanied by poor HH technique [14,15].

Because approximately 80% of HH events are performed using ABHSSs, studies of HH technique have focused on the use of ABHSSs [16,17,18]. To prevent the transmission of microorganisms between patients and dental staff, high compliance with infection control practices is required [19].

The HH technique is relatively simple. In the context of the pandemic infection with the SARS CoV-2 virus, the rules of HH, the hygiene technique, and the products to be used were covered in various forms. Even so, in practice there are still non-conformities regarding this aspect. Ignorance or lack of compliance can be reasons for non-adherence with HH techniques.

In this context, the purpose of this paper is to follow the realization of the hand washing technique with hydroalcoholic solution applied before receiving the recommendations of HH technique and after receiving the recommendations of HH technique.

The null hypothesis is that there are no differences in HH with hydroalcoholic solution between subjects before receiving technical recommendations and after receiving technical recommendations.

## 2. Materials and Methods

### 2.1. Setting and Participants

A descriptive, cross-sectional study was conducted from May 2022 to June 2022. Participants were students currently enrolled at the medical dental school in Iasi, Romania. According to the secretary of the Faculty of Dental Medicine within the UMF Gr. T. Popa (University of Medicine and Pharmacy Grigore T. Popa) from Iasi, the total population of dental medical students in the dentistry program working towards Bachelor of Dentistry in French language degrees was 89 students [20]. The calculated sample size was made with a formula for confidence level of *p* = 95%, z = 1.96, and with margin of error by 5% by population size *n* = 89 [21]. The resulting calculated sample size was 73 students. The selected sample was representative for this study. A total of 73 students were included in the study sample. Participant sampling was volunteer based.

### 2.2. Study Group

The study included dental medicine students selected from the second year dental medicine class teaching in French 2021/22 “Grigore T. Popa” University of Medicine and Pharmacy in Iasi. The selection of the study group was made following selection criteria in accordance with ethical rules and good practices of study. Ethical acceptance for this study was given in No. 184/05.05.2022. The inclusion criteria were students enrolled in the dental medicine teaching department in French degree 2021/2022 class; students attending second year; students attending preventive medicine department II classes of the preventive dentistry department; students who consented to participate in the study. The exclusion criteria were students attending in another teaching language; students in another year of study; students who did not agree to participate in the study. The students considered eligible were those who agreed to participate. A total of 73 subjects entered the study. After analyzing the images, three subjects were eliminated, leaving seventy subjects for evaluation.

### 2.3. Demographic Characteristics

The identifying variables included date, center, academic course, dental medicine group, and sex.

### 2.4. Domain: Knowledge Data

The study was conducted by the same professionals in preventive medicine on several days and on different schedules to study the whole sample of students. Small groups were established with seven students. The study was conducted in two stages. In the first stage, HH was performed without any recommendation. In the second stage, a theory lesson about the antiseptic hand rubbing was given. During practical teaching, dental medical students attended a simulated specialty dental medical practice session, and then, HH was performed following the WHO recommendations [1].

### 2.5. Materials

The study included the use of a fluorescent solution for monitoring compliance with HH. The name of the solution is Fluo-Add LOT 20063M12-0752-839973-2018-11-12., REF 180127. The solution contains water, propylene glycol, UV-Marker, and phenoxyethanol. Using a pipette, 1 mL of Fluo-Add solution was applied in 100 mL of hydroalcoholic solution prepared according to the formulation II recipe. Isopropyl alcohol 75%, glycerol 1.45%, and hydrogen peroxide 0.125% [1] are recommended by the WHO. The microbicidal activities of the two formulations recommended by the WHO were tested by a WHO reference laboratory according to EN (EN 1500) standards [1]. A fluorescent solution with a concentration of 1% was made. The Wood lamp for dermatology with 4 × 5.5 W ultraviolet tubes with a wavelength of 360 nm, dimensions 30 × 21 × 7 cm, power supply 230 V, and 50 Hz was used for highlighting. The lamp was placed in a black box with a slot at the top and an opening in the front of the box to allow the hands to be inserted. The photo camera had 12 megapixels. The image dimensions were 1024 × 768, and 1600 × 768 with a width of 1024/1600 pixels and a height of 768 pixels and a depth of 24 bits. The horizontal resolution was 96 dpi, and the vertical resolution was 96 dpi.

### 2.6. Protocol

#### 2.6.1. Stage I

The subjects were instructed to take off their jewelry, apply consultation gloves, and rub their hands with hydroalcoholic solution while performing their usual movements. The amount applied (usually 3 mL) was the one recommended in the protocol of Robinson et al. [22]. Thus, the application of the product on the hands was followed. Subjects performed their HH movements until the product evaporated. Each subject received a serial number. Subjects placed their hands in the fluorescent highlighting device. The images were taken from the front with a photo camera. The images were taken from the backhands and from the palms. The images were saved according to the serial number with the initial mention. The gloves were disposed of in the biohazard container.

#### 2.6.2. Stage II

The technique of HH according to WHO recommendations was explained to the subjects. It was explained to the subjects to perform 10 movements for each surface. The HH poster with the WHO instructions [1] to rub with a hydroalcoholic solution was presented.

The subjects put on consultation gloves. A quantity of 3 mL of solution was applied to them. Subjects performed 10 movements for each surface according to WHO recommendations [1]: (1) Apply a palmful of the product in a cupped hand, covering all surfaces; (2) Rub hands palm to palm for 10 movements; (3) Right palm over left dorsum with interlaced fingers and vice versa for 10 movements; (4) Palm to palm with fingers interlaced for 10 movements; (5) Backs of fingers to opposing palms with fingers interlocked for 10 movements; (6) Rotational rubbing of left thumb clasped in right palm and vice versa for 10 movements; (7) Rotational rubbing, backwards and forwards with clasped fingers of right hand in left palm and vice versa for 10 movements; (8) Once dry, your hands are safe. Then, depending on the serial number, images were taken for the palms and the backhands to highlight the coverage rate with fluorescent solution. The images were taken with the same camera used in stage I. The pictures were saved according to the serial number with the final mention.

### 2.7. Data Collection

The pictures were introduced for analysis in the Image Color Summarizer program—RGB and HSV Image Statistics [23]. Descriptive color statistics for an image were performed at 200 pixels resolution. The average, median, minimum, and maximum of each component of RGB, HSV, LCH, and lab were reported. Average hues were calculated using the mean of the circular quantities. The average color hue, saturation and percentage value, and which colors are most representative of the image followed. The color-clustering function evidenced the representative colors of the image and showed how the pixels in the image partition into groups. Color clusters were calculated using k-means clustering. Colors in the image were clustered into three groups. The average color of the colors for each cluster were shown. The name was the closest named color, and its distance was shown using ΔE. The tags were the set of words formed by all named neighbors within ΔE ≤ 5. The list of words above was the set of all unique words in this set of words. In the Image Cluster Partition Pixels of the image assigned to each cluster, the cluster colors were sized by the number of pixels. The coverage of the hands with fluorescent solution was evaluated by the brightest color. The border was the color of the cluster as calculated by the average value of its pixels (Figure 1 and Figure 2).

In the final calculation, the darkest image was removed, and the surface of the gloves remained, which was rated at 100%. By applying the simple rule of three, a percentage of 100% was calculated for the surface covered with fluorescent solution and a percentage of 100% for the surface not covered with fluorescent solution. Thus, after applying this algorithm, the final percentages of the covered area and the uncovered area was noted in the database. In the description of Figure 1 and Figure 2 you can follow the calculation algorithm.

The data were assigned to the groups as follows: I-C-B = initial covered backhand; F-C-B = final covered backhand; I-NC-B = initial uncovered backhand; IF-NC-B = final uncovered backhand; I-C-P = initial covered palm; F-C-B = final covered palm; I-NC-B = initial uncovered palm; IF-NC-B = final uncovered palm.

### 2.8. Statistical Data

The data was collected and introduced into a database. Excel of Microsoft Office 2007 was used for the coding of the obtained data. The cut-off point of statistical significance, *p*, was set at 0.05. The data were analyzed using IBM-SPSS version 26 (IBM, Armonk, NY, USA), and *p* ≤ 0.05 was considered statistically significant. A separate descriptive analysis of the variables was conducted, presenting the mean corresponding to the quantitative variables and centralizing measures as well as dispersion of the quantitative variables, and the non-parametric test, Independent-Samples Mann–Whitney U Test was used. A Student *t*-test Paired Samples was used for the quantitative variables and considered as significative the values *p* > 0.05.

## 3. Results

### 3.1. Demographic Data

The study sample included 70 subject students from the dental medicine faculty of UMF Gr. T. Popa, Iasi, Romania in their second year of study. The distribution by gender was 45.7% (32) female and 54.33% (38) male (Table 1).

### 3.2. Descriptive Statistics of Analyzed Data

The degree of coverage with fluorescent solution was evaluated as a percentage. Thus, an average covered area of 60.89% (±17.17), 28.84: 86.11, was finally obtained for the backhand, compared to 52.07% (±17.04), 19:85.23, initial and an average covered area of 69.91% (±13.5), 31.61:93.41, was finally obtained for the palm compared to 59.74% (±16.64), 26.13:92.72, initial (Table 2).

Frequency distribution of data obtained for covered backhand and palm initial and final are presented in Figure 3a,b and Figure 4a,b.

### 3.3. Comparative Evaluation of the Coverage Areas at the Backhands by Gender

The comparative evaluation of the coverage areas at the backhands by gender did not show significant differences for either the initial or the final evaluations. For the initial versus final evaluation for female (*n* = 32) and male (*n* = 38), mean ranks were 33.03:34.09 and 37.58:36.68, respectively (Figure 5a,b).

The comparative evaluation of the uncoverage areas at the backhands by gender did not show significant differences for either the initial or the final evaluations. For initial versus final evaluation for female (*n* = 32) and male (*n* = 38), mean ranks were 37.13:36.91 and 34.13:34.32, respectively (Figure 6a,b).

### 3.4. Evaluation of the Coverage Areas at the Palms

The comparative evaluation of the coverage areas at the palm by gender did not show significant differences for either the initial or the final evaluations. For initial versus final evaluation for female (*n* = 32) and male (*n* = 38), the mean ranks were 35.63:36.75 and 35.39:34.45, respectively (Figure 7a,b).

The comparative evaluation of the uncoverage areas at the palm by gender did not show significant differences for either the initial or the final evaluations. For initial versus final evaluation for female (*n* = 32) and male (*n* = 38), the mean ranks were 35.38:35.61 and 35.61:36.55, respectively (Figure 8a,b).

### 3.5. Comparative Analysis of the Studied Groups

Comparative analysis of the studied groups showed statistically significant directions in the final evaluation compared to the initial one for both the backhand and the palm *p* ≤ 0.005 (Table 3 and Table 4).

## 4. Discussion

The novelty of this work consists of quantifying the precise degree of coverage without limiting the assessment of some areas with certain scores. In this way, staff awareness can lead to better infection control.

Hygienic hand disinfection with an alcohol-based hand rub is the preferred treatment to be carried out after patient care activities that could lead to contamination of the hands of the health care workers. Applying an antiseptic hand-rub product to all surfaces of the hands reduce the number of microorganisms present. Many studies confirmed that the alcohol-based hand rub reduced the bacterial colonies better than the conventional hand wash products, such as chlorhexidine gluconate 4% and non-medicated soaps [24,25,26,27]. There are studies that emphasize the efficiency of hand washing with a hydro-alcoholic solution compared to liquid soap [28] and others that highlight the fact that conventional hand wash and alcohol-based hand rub show equal effectiveness in reducing the microbial load [29]. Many studies reported that compliance to alcohol hand rub increased compared to the conventional hand wash [30,31,32].

The main limitation is that it must be applied to clean hands. Another limitation is that some areas of the hands are frequently omitted. Huber et al. suggested the use of alcohol-based hand rub as a HH method in the dental health care settings when compared to the conventional hand wash because the hand rub is less time consuming and easy to apply [33].

The WHO hand wash technique recommends six steps, and the CDC recommends three steps. Chow et al. [34] found no difference in the effectiveness of the techniques, whereas Reilly et al. [35] found the WHO six-step technique to be more effective, and Tschudin-Sutter et al. [36] reported that a modified three-step technique that focused on the fingertips and thumbs was more effective than the WHO six-step technique.

It may also increase compliance and potentially improve the HH practice within the clinical setting given that suboptimal rates of HCW compliance with the WHO six-step technique have been previously reported in studies worldwide [37,38,39,40,41,42].

However, there are limits to the amount of time that can be saved with different techniques. According to the current understanding, when using ABHSs, the hands should be allowed to dry after performing the technique and before proceeding.

It is noteworthy that Reilly et al. [35] found that the efficacy of the WHO six-step technique was enhanced when it was performed with 100% accuracy (correct steps, correct order), whereas Pires et al. [40] showed that the efficacy of the WHO six-step technique was enhanced when the order of steps was changed—when the step for the fingertips, normally the last step, was performed first.

This not only raises questions about what technique is best but also suggests that techniques can be modified to enhance their effectiveness.

Routine methods for nonsurgical dental procedures are washing with plain soap or antimicrobial soap and water. However, several reports showed that switching to alcohol-based hand rub decreases time and increases compliance with any other HH method [40,41].

HH behavior varies significantly among HCWs within the same unit, institution, or country, thus suggesting that individual features could play a role in determining behavior [43,44].

Research into HH using behavioral theory has primarily focused on the individual and cognitive factors and nursing unit workload [45,46,47]

It was also highlighted that individuals’ HH behaviors are not homogeneous [44] and that doctors are usually less compliant with recommended practices for HH than are other HCWs [48].

The subjects showed compliance with HH in the sense that in practice, this technique is easy to apply. Cognitive programs aiming to modify HCWs’ HH behavior should consider adjusting the benefits to include self-protection and patient protection. The dynamic of behavioral change is complex and involves a combination of education, motivation, and system change [48].

Wide dissemination of HH guidelines alone is not sufficient motivation for a change in HH behavior [49,50,51].

With our current knowledge, it can be suggested that programs to improve HH compliance in HCWs cannot rely solely on awareness but must take into account the major barriers to altering an individual’s pre-existing HH behavior [49].

Patterns of HH behavior are developed and established in early life. As most HCWs do not begin their careers until their early twenties, improving compliance means modifying a behavior pattern that has already been practiced for decades and continues to be reinforced in community situations.

Patterns of HH represent a complex, socially entrenched, and ritualistic behavior. We can say that single interventions have failed to induce a sustained improvement in behavioral subjects, an aspect highlighted in other specialty studies [52,53].

The initial HH evaluation was performed according to the involvement of each individual in his hygiene. Therefore, the subjects performed the HH movements for self-protection reasons. The technique applied by each subject was the one practiced day by day to eliminate personal feelings of unpleasantness and discomfort.

The subjects responded positively to the training session as they were aware of the quality of HH as a priority in the dental medical act.

The benefits of a single explanation and following the correct protocol is obvious.

Thereby, the belief in the effectiveness of preventive strategies is underlined, reinforcing appropriate HH behavior.

The impact in practice of each behavioral factor influencing HH must be carefully measured and considered (WHO).

Evaluation of the effectiveness of HH guidelines or recommendations on the ultimate outcome, i.e., the HCAI rate, is certainly the most accurate way to measure the impact of improved HH.

In a study conducted on a sample of 40 hospitals in the U.S., it was highlighted that in only 44.2% of hospitals a program of implementation of HH guidelines with alcohol-based rubbing products was carried out, and the HH compliance rates were no higher than 56.6%, and in addition, the correlation of lower infection rates with higher compliance was demonstrated only for bloodstream infections [49].

Monitoring HH is very important, an aspect identified in numerous specialized studies [47,54,55,56,57,58]. However, HH performance is only one step in infection control. Excluding the structural aspects related to the quality and availability of alcoholic products, the correct application of the steps of the hand-washing technique with hydroalcoholic solution is very important. Furthermore, monitoring HH by direct methods is a way to highlight some deficiencies in its implementation. Detection of HH compliance by a validated observer (direct observation) is currently considered the gold standard in HH compliance monitoring [6]. An issue would be related to how we can make medical staff aware that the steps of the technique must be followed in a sustained manner. The visual image of the efficiency of HH is a positive aspect that can motivate the staff in applying the correct technique.

Concerns about HH with hydroalcoholic products as well as the quest for less expensive monitoring approaches have shaped the consumption of HH products [59]. Moreover, the methods based on product consumption cannot determine if the technique for HH is correct. The application technique has not been standardized throughout the world. The WHO approach for hand rubbing with an alcohol-based solution requires six basic steps for hygienic hand antisepsis. One study demonstrated that keeping the hands wet with the rub is more important than the volume used [60].

The size of the hands ultimately determines the volume required to keep the skin area wet during the entire time of the hand rub. The application technique has not been standardized worldwide. Our study showed that wetting surfaces should be performed with hand rubbing movements.

Causes of potential bias arising from HH direct observation are observation, observer, and selection bias. Thus, we set out to follow the technique visually by recording images.

The recommendations for using the amount of hydroalcoholic solution for the manufacturer are between two and three milliliters. In addition, the size of the hands ultimately determines the volume needed to keep the skin area moist throughout the rubbing of the hands. In this study we used three milliliters of solution for HH with hydroalcoholic solution. A bias is the insufficient sample size, representing a major threat to meaningful monitoring outputs. The method works well on images with relatively well-defined color boundaries and does not work well on images with smooth gradients that transition across a large range of colors (in hue, brightness, and saturation). Additionally, other factors that could influence the results would be the resolution of the camera, the identification of the shades on the surface of the hands, and the slightly bent position on the fingers. Another limitation of this study would be that the subjects applied another technique as soon as they received information and under the direct guidance of the work team. Through this study, we wanted to show that an improvement in the quality of hand hygiene can be achieved. That is why it is imperative that subjects be alerted as much as possible regarding this aspect. Since after the immediate training the surfaces of the hands were not covered 100%, it can be emphasized that many educational steps are needed. In this context, a limitation of the study would be the loss of notions over time and the return to old habits. The future implications of this study will help raise the awareness of the medical staff, educate the medical staff, and perhaps change their attitudes towards hand hygiene, ensuring better control of infections. The quantification of the values and the exact imaging demonstration of the efficiency can motivate the practitioners more in changing the behaviors. Thus, future studies are needed to verify the acquisition of knowledge and the modification of attitudes regarding hand hygiene.

## 5. Conclusions

Within the limits of this study, we can draw the following conclusions. Comparative analysis of the studied groups showed statistically significant differences of the final evaluations compared to the initial ones for both the backhand and the palm. In addition, the comparative evaluations of the coverage areas at the backhands and the palm by gender did not show significant differences for either the initial or the final evaluations.

## Figures and Tables

**Figure 1 medicina-58-01108-f001:**
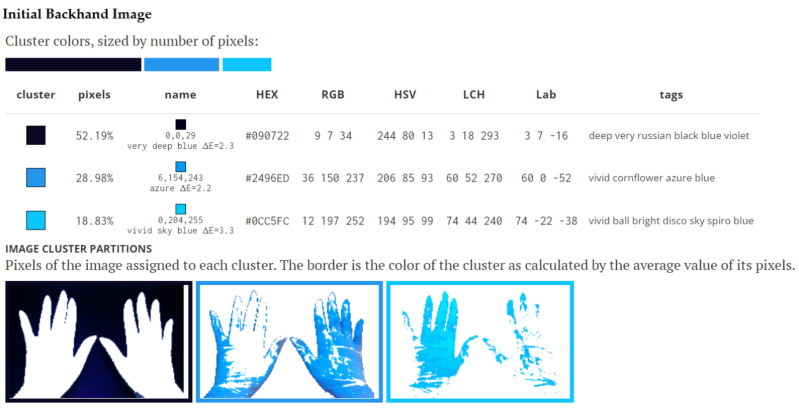
Cluster colors, sized by number of pixels: Image Cluster Partitions Pixels of the image. The image represents initial backhand image. The coverage of the hands with fluorescent solution was evaluated by the brightest color, (bright) respectively, of 18.83% compared to the uncovered surface of 28.98%. In the final calculation, the darkest image of 52.19% was removed. By applying the simple rule of three, a percentage of 100% was calculated for the surface covered with fluorescent solution and a percentage of 100% for the surface not covered with fluorescent solution. Thus, after applying this algorithm, the final percentage of covered area was 39.39% and the uncovered area was 60.61%.

**Figure 2 medicina-58-01108-f002:**
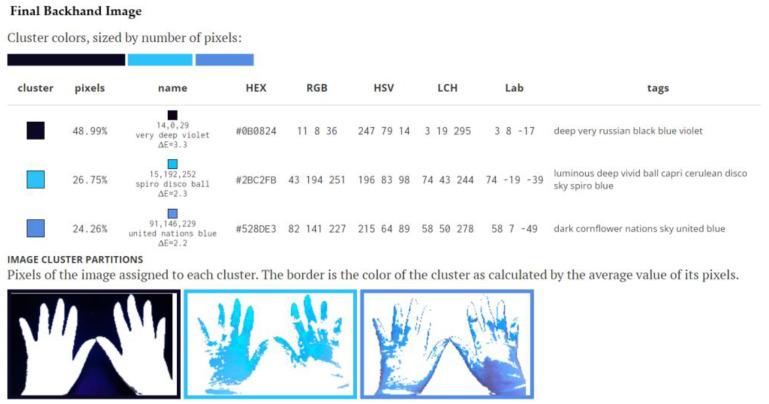
Cluster colors, sized by number of pixels: Image Cluster Partitions Pixels of the image. The image represents final backhand image. The border is the color of the cluster as calculated by the average value of its pixels. The coverage of the hands with fluorescent solution was evaluated by the brightest color, (bright) respectively, of 26.75% compared to the uncovered surface of 24.26%. In the final calculation, the darkest image of 48.99% was removed. By applying the simple rule of three, a percentage of 100% was calculated for the surface covered with fluorescent solution and a percentage of 100% for the surface not covered with fluorescent solution. Thus, after applying this algorithm, the final percentage of covered area was 52.44% and the uncovered area was 47.66%.

**Figure 3 medicina-58-01108-f003:**
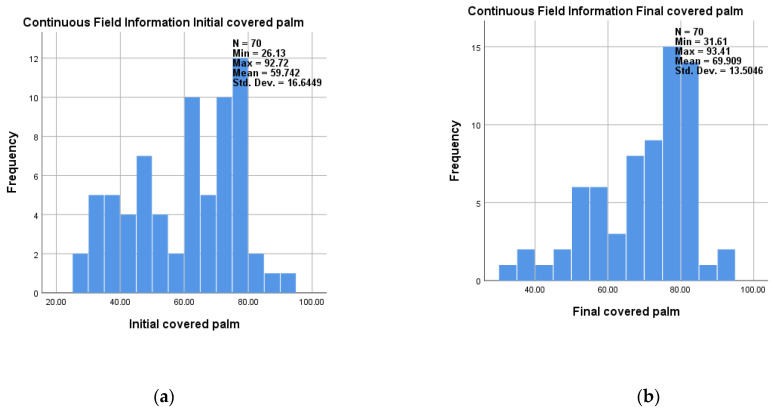
Frequency of data obtained for covered palm: (**a**) initial; (**b**) final.

**Figure 4 medicina-58-01108-f004:**
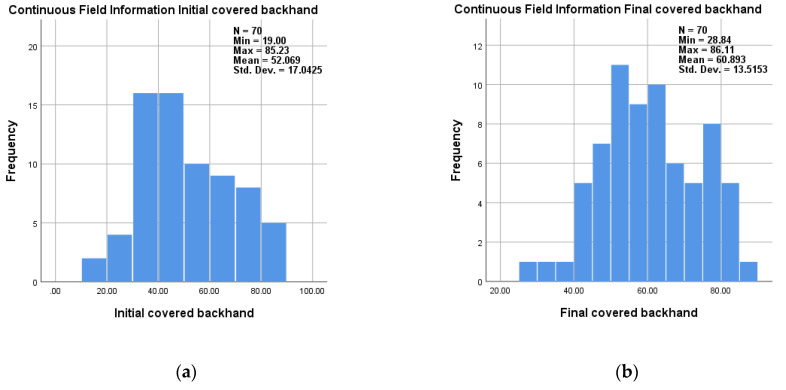
Frequency of data obtained for covered backhand: (**a**) initial; (**b**) final.

**Figure 5 medicina-58-01108-f005:**
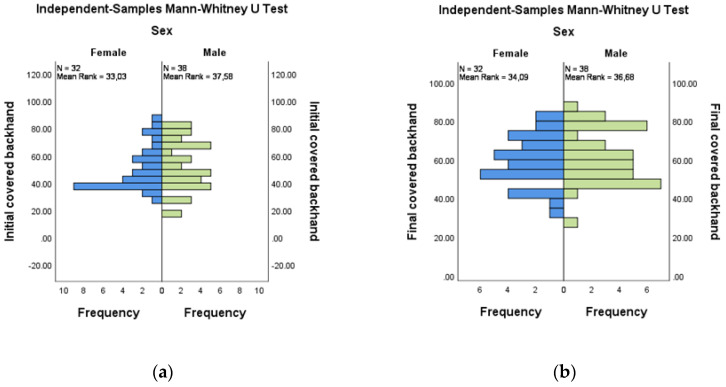
Independent-Samples Mann–Whitney U Test—covered backhand across sex: (**a**) initial; (**b**) final.

**Figure 6 medicina-58-01108-f006:**
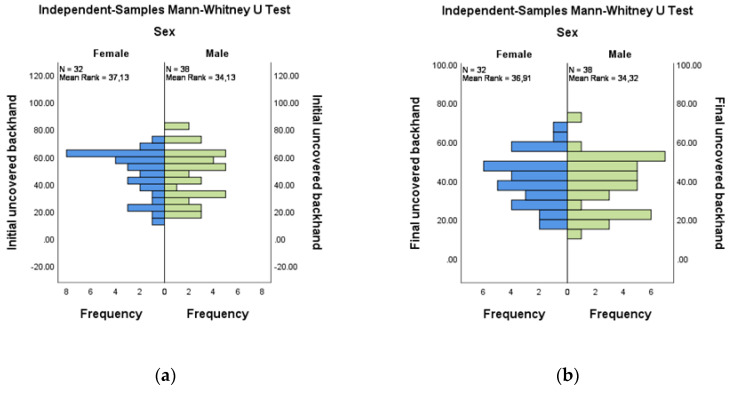
Independent-Samples Mann-Whitney U Test—uncovered backhand across sex (**a**) Initial; (**b**) Final.

**Figure 7 medicina-58-01108-f007:**
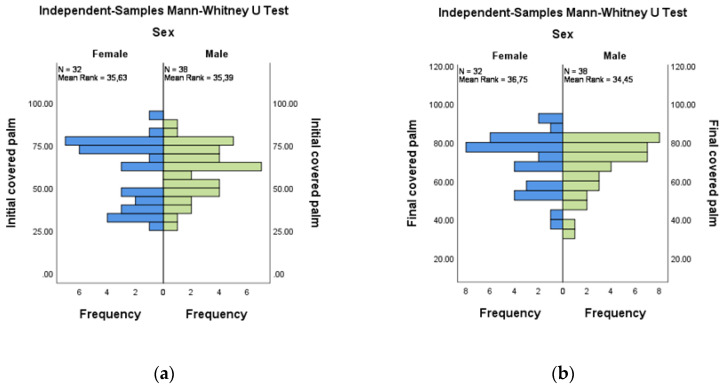
Independent-Samples Mann–Whitney U Test—covered palm across sex: (**a**) initial; (**b**) final.

**Figure 8 medicina-58-01108-f008:**
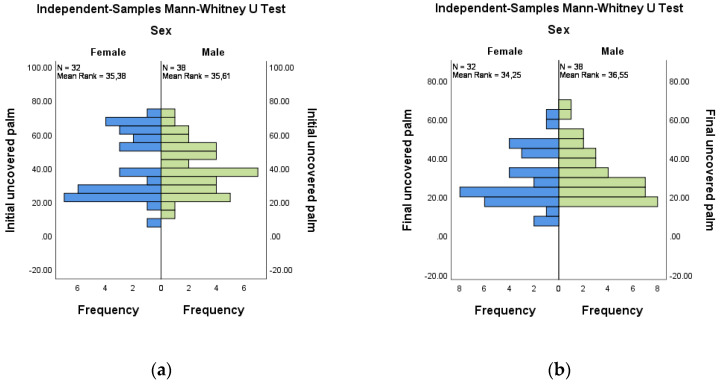
Independent-Samples Mann–Whitney U Test—uncovered palm across sex: (**a**) initial; (**b**) final.

**Table 1 medicina-58-01108-t001:** Distribution of subjects according to sex.

Sex					
		Frequency	Percent	Valid Percent	Cumulative Percent
Gender	Female	32	45.7	45.7	45.7
	Male	38	54.3	54.3	100
	Total	70	100	100	

**Table 2 medicina-58-01108-t002:** Descriptive statistics.

Descriptive Statistics					
	*n*	Minimum	Maximum	Mean	Std. Deviation
I-C-B	70	19	85.23	52.07	17.04
I-NC-B	70	14.77	81	47.34	17.17
F-C-B	70	28.84	86.11	60.89	13.51
F-NC-B	70	13.89	71.16	39.11	13.51
I-C-P	70	26.13	92.72	59.74	16.64
I-NC-P	70	7.28	73.87	40.26	16.64
F-C-P	70	31.61	93.41	69.91	13.50
F-NC-P	70	6.59	68.39	30.09	13.50
Valid N (listwise)	70				

I-C-B = initial covered backhand; F-C-B = final covered backhand; I-NC-B = initial uncovered backhand; F-NC-B = final uncovered backhand; I-C-P = initial covered palm; F-C-B = final covered palm; I-NC-B = initial uncovered palm; IF-NC-B = final uncovered palm.

**Table 3 medicina-58-01108-t003:** *t*-Test-Paired Samples Statistics and Correlations.

Pair		Mean	N	Std. Deviation	Std. Error Mean	Correlation	Sig.
1	I-C-B	52.0693	70	17.04245	2.03696	0.509	0.000 *
	F-C-B	60.8925	70	13.51530	1.61539		
2	I-NC-B	47.3386	70	17.17381	2.05266	0.492	0.000 *
	F-NC-B	39.1075	70	13.51530	1.61539		
3	I-C-P	59.7417	70	16.64494	1.98945	0.336	0.005 *
	F-C-P	69.9089	70	13.50459	1.61411		
4	I-NC-P	40.2583	70	16.64494	1.98945	0.336	0.005 *
	F-NC-P	30.0911	70	13.50459	1.61411		

I-C-B = initial covered backhand; F-C-B = final covered backhand; I-NC-B = initial uncovered backhand; F-NC-B = final uncovered backhand; I-C-P = initial covered palm; F-C-B = final covered palm; I-NC-B = initial uncovered palm; IF-NC-B = final uncovered palm; * significance level.

**Table 4 medicina-58-01108-t004:** Independent-Samples Mann–Whitney U Test Summary.

	I-C-B	F-C-B	I-NC-B	F-NC-B	I-C-P	F-C-P	I-NC-P	F-NC-P
Total N	70	70	70	70	70	70	70	70
Mann-Whitney U	687.00	653.00	556.00	563.00	604.00	568.00	612.00	648.00
Wilcoxon W	1428.00	1394.00	1297.00	1304.00	1345.00	1309.00	1353.00	1389.00
Test Statistic	687.00	653.00	556.00	563.00	604.00	568.00	612.00	648.00
Standard Error	84.82	84.82	84.82	84.82	84.82	84.82	84.82	84.82
Standardized Test Statistic	0.93	0.53	−0.61	−0.53	−0.05	−0.47	0.05	0.47
Asymptotic Sig. (2-sided test)	0.35	0.60	0.54	0.60	0.96	0.64	0.96	0.64

I-C-B = initial covered backhand; F-C-B = final covered backhand; I-NC-B = initial uncovered backhand; F-NC-B = final uncovered backhand; I-C-P = initial covered palm; F-C-B = final covered palm; I-NC-B = initial uncovered palm; IF-NC-B = final uncovered palm.

## Data Availability

The data that support the findings of this study are available on request from the corresponding author.

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
