# Peer review of "Evaluation of the Efficiency of Hand Hygiene Technique with Hydroalcoholic Solution by Image Color Summarize"

_medicina, 2022, doi:10.3390/medicina58081108_

Round 1

Reviewer 1 Report

Thank you for sharing this manuscript. Evaluation of the efficiency of hand hygiene technique with hydroalcoholic solution by Image Color Summarize.

The fact remains that maintaining hand hygiene using gold standards and proper technique is best. This has been demonstrated several times in various settings previously among different categories of study participants. Therefore, I do not see any novel findings or concepts in the present communication. Moreover, The main conclusion was- Line 416-417 :  “The comparative evaluation of the coverage areas at the backhands by gender did not show significant differences for either the initial or the final evaluations.”

What was the hypothesis to find any difference by gender? The authors have not explained, how this method can be useful in future practice or if this can be used at all? What are the future implications of this study?

The manuscript has several flaws. To give a few examples:

The authors have not given references for some of the prominent statements or methods used for example in Line 160-163- which posters were displayed?

What was the reason for using the ’10 steps’ of HH instead of the 6 steps recommended by the WHO? WHO advocates maintaining uniformity in HH technique so that it is universal and the healthcare worker can easily adapt it and apply it globally.

Author Response

Response to the review,

Thank you very much for your kindness in evaluating this article, based on your personal experience. Thanks for the pertinent comments. I hope I answered according to the specifications made. I tried to organize the answers according to your comments, as follows:

  1. This has been demonstrated several times in various settings previously among different categories of study participants. Therefore, I do not see any novel findings or concepts in the present communication.

Answer 1

The novelty of this work consists in quantifying the degree of coverage more precisely than by other methods where areas with certain scores are described and where there are greater limitations for assessing the quality of hand hygiene. In addition, this study is necessary like many others regarding hand hygiene. The existence of the guidelines does not motivate the medical staff in their attitude regarding hand hygiene. Instead, raising an alarm signal can supplement the awareness of specialists. Staff awareness will lead to better infection control.

Through this quantification, the medical staff sees, becomes aware and can change their attitude.

I added in the text, at the beginning of the discussion chapter this paragraph:

” The novelty of this work consists in quantifying the precise degree of coverage without limiting the assessment of some areas with certain scores. In this way, staff awareness can lead to better infection control.”

  1. Moreover, the main conclusion was- Line 416-417: “The comparative evaluation of the coverage areas at the backhands by gender did not show significant differences for either the initial or the final evaluations.”

Answer 2

In general, female subjects pay more attention to details. The evaluation of the technique according to gender shows that in this case the subjects used the level of knowledge. The fact that there are no differences shows that the female and male subjects have acquired the same technique and attitude.

I changed the order of the conclusions, which are as follows:

"Within the limits of this study, we can draw the following conclusions. Comparative analysis of the studied groups showed statistically significant differences of the final evaluation compared to the initial one for both the backhand and the palm. In addition, the comparative evaluation of the coverage areas at the backhands and also for the palm by gender did not show significant differences for either the initial or the final evaluations.

  1. "What was the hypothesis to find any difference by gender? The authors have not explained, how this method can be useful in future practice or if this can be used at all? What are the future implications of this study?

Answer 3

The hypothesis was between the initial and final evaluation. I changed to conclusions. I have plus and notice that there were no differences according to gender. Thanks.

I added this paragraph to the text Line 424-428

The future implications of this study will help raise the awareness of the medical staff, educate the medical staff and perhaps change their attitude towards hand hygiene, ensuring a better control of infections. The quantification of the values and the exact imaging demonstration of the efficiency can motivate the practitioners more in changing the behaviors.

  1. The manuscript has several flaws. To give a few examples:The authors have not given references for some of the prominent statements or methods used for example in Line 160-163- which posters were displayed?

Answer 4.

The posters from the WHO recommendation guide were used - I added the reference. Thanks!

  1. What was the reason for using the ’10 steps’ of HH instead of the 6 steps recommended by the WHO? WHO advocates maintaining uniformity in HH technique so that it is universal and the healthcare worker can easily adapt it and apply it globally?

Answer 5.

In this study WHO recommendations were strictly followed. Six steps were used and 10 sanitizing movements were performed for each step. It is written in paragraphs line 166. Line 168. Line 169, line 170.

In addition, I made the following changes to this manuscript.

I added the following paragraph.

A limitation of this study would be that the subjects applied another technique as soon as they received information and under the direct guidance of the work team. Through this study, we wanted to show that an improvement in the quality of hand hygiene can be achieved. That is why it is imperative that subjects be alerted as much as possible regarding this aspect. Since after the immediate training the surface of the hands was not covered 100%, it can be emphasized that many educational steps are needed. In this context, a limitation of the study would be the loss of notions over time and the return to old habits. Thus, future studies are needed to verify the acquisition of knowledge and the modification of attitudes regarding hand hygiene

I changed the word ignorance line 15 with lack of knowledge.

Thank you once again for the comments that improved the manuscript and for the time allocated to this evaluation.

Best regards,

Assoc. Professor, MD, PhD, Iulia Saveanu

Reviewer 2 Report

Thank you for sending me this manuscript to review.  There is a plethora of literature on compliance with HH guidelines.  Less so on HH technique.  

While the paper has some merit, it could be strengthened by: 

1. Data collection section: provide a rationale/discussion on why the participants wore gloves in Phase 2 but not Phase 1.  The difference between wearing and not wearing gloves would plausibly contribute to significant differences found between phases.  It needs to be discussed and a rationale provided as otherwise readers may misinterpret the results and conclude the AHR was more effective than it was.  

2. In the discussion section or the conclusion section, it should be pointed out that data for phase two were collected very shortly after the instructions/educational programme on how to use AHR.  It seems plausible again that there would be a difference found between phases given the short period of time between phases.  Did the authors consider repeating data collection while allowing a longer period of time to elapse before estimating the effectiveness of the instructions/educational programme?  Having such a short period of time between each phase could simply be attributed to the Hawthorne effect rather than any other behavioural change.  This should be discussed as it is another methodological limitation.  

3.  Can the word 'ignorance' in line 15 (abstract) be replaced with 'lack of knowledge'?  Ignorance suggests a judgement.  Lack of knowledge is the more widely used phrase in this context.  

Author Response

Response to the Review,

Thank you very much for taking the time to write this review and for the relevant recommendations. Based on your professional experience, I took into account all the highlighted aspects. Taking into account your remarks, I answered as follows:

Q1. Data collection section: provide a rationale/discussion on why the participants wore gloves in Phase 2 but not Phase 1.  The difference between wearing and not wearing gloves would plausibly contribute to significant differences found between phases.  It needs to be discussed and a rationale provided as otherwise readers may misinterpret the results and conclude the AHR was more effective than it was.  

Answer 1: The subjects wore gloves both in stage I and stage II, but after stage 1 they removed them and put on other gloves for stage II.

Write in the work at stage II line 162 that "Then the subjects put on consultation gloves". I gave a new paragraph entry to be more obvious and I cut the word ”then”.

  1. In the discussion section or the conclusion section, it should be pointed out that data for phase two were collected very shortly after the instructions/educational programme on how to use AHR.  It seems plausible again that there would be a difference found between phases given the short period of time between phases.  Did the authors consider repeating data collection while allowing a longer period of time to elapse before estimating the effectiveness of the instructions/educational programme?  Having such a short period of time between each phase could simply be attributed to the Hawthorne effect rather than any other behavioral change.  This should be discussed as it is another methodological limitation.  

Answer 2. Thank you for your attention.

Stages I and II followed each other. We did not aim to change the behavior that is practically achieved over time. We wanted to highlight the fact that if you want, you can achieve a quantifiable improvement. If every practitioner were careful and checked their attitude, a better infection control could be achieved, in general and in dental offices in particular. I specified the fact that other future studies are needed to highlight the possible differences.

I added the following paragraph.

A limitation of this study would be that the subjects applied another technique as soon as they received information and under the direct guidance of the work team. Through this study, we wanted to show that an improvement in the quality of hand hygiene can be achieved. That is why it is imperative that subjects be alerted as much as possible regarding this aspect. Since after the immediate training the surface of the hands was not covered 100%, it can be emphasized that many educational steps are needed. In this context, a limitation of the study would be the loss of notions over time and the return to old habits. Thus, future studies are needed to verify the acquisition of knowledge and the modification of attitudes regarding hand hygiene

  1. Can the word 'ignorance' in line 15 (abstract) be replaced with 'lack of knowledge'?  Ignorance suggests a judgement.  Lack of knowledge is the more widely used phrase in this context.  

Answer 3:

Thanks for the suggestion. I replaced the word ignorant with lack of knowledge.

I used the word ignorance because future professionals in the field must be aware and compliant with medical documents and ensure infection control. In the field of hand hygiene, lack of knowledge should not exist, but unfortunately it does. Sometimes the subjects know the technique but do not apply it correctly due to convenience or other reasons.

In addition,

I added in the text, at the beginning of the discussion chapter this paragraph:

” The novelty of this work consists in quantifying the precise degree of coverage without limiting the assessment of some areas with certain scores. In this way, staff awareness can lead to better infection control.”

I changed the order of the conclusions, which are as follows:

"Within the limits of this study, we can draw the following conclusions. Comparative analysis of the studied groups showed statistically significant differences of the final evaluation compared to the initial one for both the backhand and the palm. In addition the comparative evaluation of the coverage areas at the backhands and also for the palm by gender did not show significant differences for either the initial or the final evaluations.

I added this paragraph to the text Line 424-428

The future implications of this study will help raise the awareness of the medical staff, educate the medical staff and perhaps change their attitude towards hand hygiene, ensuring a better control of infections. The quantification of the values and the exact imaging demonstration of the efficiency can motivate the practitioners more in changing the behaviors.

I put the bibliographic source regarding the poster. Line 163

Thank you once again for the comments that improved the manuscript and for the time allocated to this evaluation.

Best regards,

Assoc. Professor, MD, PhD, Iulia Saveanu

Round 2

Reviewer 1 Report

The changes and modifications made in text are acknowledged. it has uplifted the content, importance and readability of the manuscript.

It is suggested to go through the modified parts and change the aim of the study as explained in the response to reviewer's comment that, This study will help to motivate the health care workers towards techniques of hand hygiene.

This is an important point. Therefore, It is also suggested to highlight this point in discussion as well. 

Reviewer 2 Report

Dear all, 

Many thanks for sending me the revised manuscript for review.  

I think it reads much better now.  Your revisions have provided detail/discussion for the reader which are very useful.

Well done.